# Acceptability of short message service reminders as the support tool for PrEP adherence among young women in Mukono district, Uganda

**Muteebwa Laban**[1]*, **Edith Joloba Nakku**[2], **Joan Nangendo**[1], **Dan Muramuzi**[2], **Faith Akello**[1], **Sabrina Kitaka Bakeera**[3], **Fred Collins Semitala**[4], **Aggrey S. Semeere**[5], **Charles Karamagi**[1]

1 Clinical Epidemiology Unit, School of Medicine, College of Health Sciences, Makerere University, Kampala, Uganda, 2 Department of Epidemiology and Biostatistics, School of Public Health, College of Health Sciences, Makerere University, Kampala, Uganda, 3 Department of Pediatrics and Child health, School of Medicine, College of Health Sciences, Makerere University, Kampala, Uganda, 4 Makerere University Joint AIDS Program, College of Health Sciences, Makerere University, Kampala, Uganda, 5 Department of Prevention, Care and Treatment, Infectious Diseases Institute, College of Health Sciences, Makerere University, Kampala, Uganda

* mutlabans@gmail.com

**Data Availability Statement:** The data underlying the study findings can be accessed through the public repository at DOI: 10.5061/dryad.cvdncjt8h.

## Abstract

Adolescent girls and young women (AGYW) have a disproportionately high incidence of HIV compared to males of the same age in Uganda. AGYW are a priority sub-group for daily oral Pre-Exposure Prophylaxis (PrEP), but their adherence has consistently remained low. Short Message Service (SMS) reminders could improve adherence to PrEP in AGYW. However, there is paucity of literature about acceptability of SMS reminders among AGYW using PrEP. We assessed the level of acceptability of SMS reminders as a PrEP adherence support tool and the associated factors, among AGYW in Mukono district, Central Uganda. We consecutively enrolled AGYW using PrEP in Mukono district in a cross sectional study. A structured pre-tested questionnaire was administered to participants by three trained research assistants. Data were analyzed in STATA 17.0; continuous variables were summarized using median and interquartile range (IQR) while categorical variables were summarized using frequencies and percentages. Acceptability of SMS was defined as willingness to accept SMS reminders to support PrEP adherence and was assessed using the seven constructs of the theoretical framework of acceptability. The relationship between the outcome and independent variables was assessed using a modified Poisson regression with robust standard errors. During the month of August 2022, 142 AGYW with median age 22 years (IQR; 18, 24) of whom 80.3% owned a personal mobile phone were assessed. SMS reminders were highly acceptable [90.9%, 95% Confidence Interval (CI) [84.9, 95.0]]. Rural residence was negatively associated with acceptability of SMS reminders (aPR: 0.92, 95% CI (0.84, 0.99)) and having belief that SMS cannot breach individual's privacy (aPR: 1.40, 95% CI (1.07, 1.84)) was positively associated with acceptability of SMS reminders. The acceptability of SMS reminders was high in this sub-population. SMS reminder can be

**Funding:** The Research reported in this publication was supported by the Fogarty International Center and the National Institute on Mental Health of the National Institutes of Health under Award Number D43 TW010037. The content is solely the responsibility of the authors and does not necessarily represent the official views of the National Institutes of Health. The funders had no role in the study design, data collection and analysis, decision to publish or preparation of the Manuscript. FCS received the funding for this work.

**Competing interests:** The authors have declared that no competing interests exist.

leveraged to support AGYW to adhere to PrEP but should be designed in a way that maintains confidentiality, and supports AGYW living in rural settings.

## Background

Globally, nearly 5,000 adolescent girls and young women (AGYW) are newly infected with HIV every week [1]. In Uganda, AGYW contributed about a third of new HIV infections in 2020, despite only representing 10% of the population [2]. AGYW are one of the most at risk populations in Uganda and are therefore priority beneficiaries for HIV Pre-Exposure Prophylaxis (PrEP) services [3]. However, various studies have reported that AGYW have low levels of adherence to oral PrEP [4, 5] which reduces its efficacy to prevent HIV infection. For this reason, the 2016 World Health Organization (WHO) technical guidance on PrEP use highlighted the need for research on interventions aimed at supporting AGYW to adhere to PrEP [6].

Short Message Service (SMS) reminders sent on individuals' mobile phones have been shown to support adherence to PrEP among men who have sex with men (MSM) [7]. SMS reminders have also been shown to improve treatment adherence among people living with HIV (PLHIV) on ART] [8, 9], tuberculosis patients on anti-tuberculosis drugs [10] and other patients on long-term treatments [11]. However, there is paucity of literature on whether this evidence based intervention (EBI) is acceptable to AGYW using PrEP in Uganda. The SMS reminders can be one-way (from provider to the recipient) or two-way (that allows exchange of information between the provider and recipient). Moreover, most studies have measured acceptability of SMS reminders, using a single question measure which may carry a lot of bias. Measuring this complex attribute using a composite question approach like utilizing the Theoretical Framework of Acceptability (TFA) may provide valid estimates of acceptability. The TFA has seven constructs that all measures important components that influence the overall acceptability of such a technology-based intervention [12]. Acceptability of SMS reminders can be modified by factors like age, area of residence, education level, mobility patterns, and access to cell phones. A study done by Rana and colleagues in Uganda on HIV patients on ART reported that perceived breach of confidentiality and perceived stigma influenced the acceptability of SMS reminders [13]. AGYW using PrEP are likely to have increased levels of perceived stigma because they are in a developmental transition and in a complex family web [14], and this may influence their willingness to accept SMS reminders.

Mukono is a peri-urban district in Central Uganda, where 70% of females aged 18–30 years and 54.2% of females aged 10 years and above own a personal mobile phone according to the 2014 census [15], and the coverage could have increased presently. This high coverage of mobile phones can be leveraged to implement the SMS reminder as a strategy to improve adherence to oral PrEP. Therefore, this study sought to estimate the acceptability of SMS reminders and the associated factors among AGYW using oral PrEP in Mukono district.

## Methods

### Study design and setting

We conducted a cross sectional study among AGYW using PrEP in Mukono, a district which is about 21 Km East of Kampala city, the capital city of Uganda. The district has a population of 720,100 people of whom about 371600 are females [16]. The HIV prevalence among persons aged 15 years and above in Mukono district as of 2021 is 6.1% which is higher than the

national prevalence (5.4%) thus is a priority district for HIV prevention [17]. The President's Emergency Plan for AIDS Relief (PEPFAR) program provides HIV prevention services including PrEP services in Mukono district through the Makerere University Walter Reed Project (MUWRP). MUWRP implements PEPFAR's "Determined, Resilient, Empowered, AIDS-free, Mentored and Safe" (DREAMS) project across the district with an overarching goal of reducing the incidence of HIV among AGYW through provision of health services, enhancing social skills, economic empowerment and business start-ups, AGYW in this program access HIV prevention services including PrEP. This program also provides safe spaces (sites that provide privacy) to AGYW where they often meet and receive HIV prevention services, skilling, socio-economic empowerment and peer-to-peer support.

## Study population, inclusion and exclusion criteria

We recruited 142 AGYW who were current participants in the PEPFAR DREAMS project. Participants were eligible if they were: (1) aged 15–24 years; (2) at least one month since the start of oral PrEP; (3) gave written informed consent (for adults and emancipated or mature minors) or assent for those less than 18 years (and not classified as emancipated or mature minors) in addition to written informed consent of their guardians/parents.

AGYW who were not found at the study site during the time of data collection and those who were incapacitated to complete the study procedures due to sickness, or any other reason were excluded from the study.

## Sampling and data collection procedure

The AGYW in the DREAMS project met at the designated safe spaces at least once a week for project-related activities. The study team visited all the safe spaces on the meeting day of each of the groups and AGYW with PrEP prescriptions were invited to participate in the study through their peer-leaders. The questionnaire was administered by trained research assistants to consecutively enrolled AGYW until the required sample size was accrued. The study was conducted between 1st and 30th August 2023.

## Study variables

The dependent variable was acceptability of SMS reminders defined as willingness of an individual to use SMS reminders as a PrEP adherence support tool, if the services were to be provided. Acceptability of SMS reminders was measured using the seven constructs of the theoretical framework of acceptability (TFA) [12]. The constructs include affective attitude, burden, perceived effectiveness, ethicality, intervention coherence, opportunity costs, and self-efficacy. Each construct was assessed using one five-level Likert item question with responses weighted from one to five, where the least weight represent low acceptability and highest weight represent high acceptability. The summated score of weights from all the seven questions was dichotomized using the median of the possible scores that range from 7 to 35 (median (50th percentile) was 21). Participants with a score above 21 were categorized as having accepted SMS reminders. The use of percentiles to categorize continuous outcomes of acceptability has been used previously in literature [18].

The independent variables included; socio-demographics like age, area of residence, religion, marital status, education level, current schooling status and individual factors like possession of a personal mobile phone, use of phone alarm (in past28 days), mobility (in past28 days), disclosure of PrEP use, period since PrEP initiation, missed doses (in past28 days), literacy on use of SMS features and belief that SMS can breach an individual's privacy.

## Data analysis

Categorical variables were summarized using frequencies and percentages while continuous variables were summarized using the median and inter-quartile range (IQR). The level of acceptability was the primary outcome, and was calculated as a proportion of total participants that accepted the use of SMS reminders and its exact confidence interval (CI) was computed. The level of acceptability among participants who currently had a mobile phone and those who didn't have was compared using a Chi-square test. We used the modified Poisson regression with robust standard errors to assess for the association between the dependent and independent variables. At unadjusted analysis, independent variables with a P-value of less than 0.2 were included in the multivariate model. At adjusted analysis, interaction was assessed for by using a chunk test (-2 log-likelihood ratio test). Confounding was assessed for, where an independent variable that changed the adjusted prevalence ratio (aPR) of another variable in the model by a magnitude of more than 10% was considered a confounder. Independent variables with a P value ≤0.05 were considered statistically significant in the final multivariate model. We analyzed data in STATA version 17.0 (Texas, USA)

## Ethical approval

The study was reviewed and approved by the Makerere University School of Medicine Research Ethics Committee under protocol number Mak-SOMREC-2022-343. Administrative clearance was sought from MUWRP and the office of the district health officer-Mukono district. Adult participants and emancipated/mature minors independently gave written informed consent whereas minors gave written informed assent in addition to the written consent of their parents/guardians. Participants were compensated with 10,000 Uganda shillings for their time.

# Results

## Socio-demographic characteristics of study participants

Over a one month period (August, 2022) we enrolled 142 participants with median age of 22 years (IQR: 18, 24). Majority of participants lived in rural setting (71.8% (102/142)) and slightly more than half had stayed in the current area of residence for less than 5 years (59.1% (84/142)). More than half of the participants self-identified as married (55.6% (79/142)); had attended post-primary education (54.2% (77/142)), and half of the participants were unemployed (50.7% (72/142)). About 28.2% (40/142) of the participants were affiliated to the Catholic religion, while 26.8% (38/142) were affiliated to the Anglican religion.

Majority of study participants were not in school (71.8% (102/142)). More than three quarters reported not to have moved away from home for two consecutive days in the past four weeks (80.3% (114/142)). More than three quarters of study participants owned a mobile phone (80.3% (114/142)). Of those who owned a mobile phone, almost a third reported that they share it with another person (35.1% (40/114)). Only 8.4% (12/142) couldn't use the SMS features of a mobile phone (Table 1).

## Individual characteristics of study participants

About half of the study participants had initiated PrEP in the past three or more months (47.2% (67/142)). Slightly more than half (56.3% (80/142)) reported to have disclosed PrEP use to another person, with a third disclosing to their parents (37.1% (23/62)). Of those who disclosed their PrEP use, only 4.8% (3/62) reported getting no support from the person they disclosed to. A quarter of study participants reported to have missed at least one PrEP dose in the

**Table 1. Socio-demographic characteristics of AGYW aged 15–24 years using PrEP in Mukono district.**

| Variable | Categories | Frequency (N = 142) | Percentage (%) |
|---|---|---|---|
| **Age** | | | |
| | 15–19 Years | 50 | 35.2 |
| | 20–24 Years | 92 | 64.8 |
| **Period of stay in the current area of residence** | | | |
| | 5Years and above | 58 | 40.9 |
| | Less than 5 Years | 84 | 59.1 |
| **Residence** | | | |
| | Urban | 40 | 28.2 |
| | Rural | 102 | 71.8 |
| **Marital status** | | | |
| | Married/Cohabiting | 79 | 55.6 |
| | Single | 63 | 44.4 |
| **Religion** | | | |
| | Anglican | 38 | 26.8 |
| | Catholic | 40 | 28.2 |
| | Moslem | 36 | 25.3 |
| | Others* | 28 | 19.7 |
| **Occupation** | | | |
| | Employed | 70 | 49.3 |
| | Unemployed | 72 | 50.7 |
| **Education level completed** | | | |
| | Primary level or less | 65 | 45.8 |
| | Post-Primary level | 77 | 54.2 |
| **Current schooling status** | | | |
| | Schooling | 40 | 28.2 |
| | Not schooling | 102 | 71.8 |
| **Moved away from home for 2 consecutive days in past 4 weeks** | | | |
| | Yes | 28 | 19.7 |
| | No | 114 | 80.3 |
| **Ever had a personal mobile phone** | | | |
| | Yes | 120 | 84.5 |
| | No | 22 | 15.5 |
| **Currently own a personal mobile phone** | | | |
| | Yes | 114 | 80.3 |
| | No | 28 | 19.7 |
| **Share Mobile phone with another person$** | | | |
| | Yes | 40 | 35.1 |
| | No | 74 | 64.9 |
| **Literacy on use of SMS features of a mobile phone** | | | |
| | Literate | 130 | 91.6 |
| | Illiterate | 12 | 8.4 |

*Born again, Seventh Day Adventist, $n = 114

past 4 weeks (26.1% (37/142)), mostly due to forgetting to take the PrEP pill (48.7% (18/37)). Majority of the study participants reported to have read or sent an SMS in the past 4 weeks and about half reported to have used a mobile phone alarm (47.9% (68/142)) in the past four weeks (Table 2).

**Table 2. Individual characteristics of AGYW aged 15–24 years using PrEP in Mukono district.**

| Variable | Categories | Frequency (N = 142) | Percentage (%) |
|---|---|---|---|
| **Period since PrEP Initiation** | | | |
| | Past one month | 49 | 34.5 |
| | Past two months | 26 | 18.3 |
| | Past three months or more | 67 | 47.2 |
| **PrEP visits in the past 3 months** | | | |
| | None | 30 | 21.1 |
| | Once | 52 | 36.6 |
| | Twice | 31 | 21.9 |
| | Three or more times | 29 | 20.4 |
| **Disclosure of PrEP use** | | | |
| | Yes | 62 | 43.7 |
| | No | 80 | 56.3 |
| **Individual to whom participant disclosed PrEP Use to*** | | | |
| | Parent | 23 | 37.1 |
| | Spouse/Sexual partner | 19 | 30.6 |
| | Peer/Friend | 20 | 32.3 |
| **Support to continue taking PrEP following disclosure*** | | | |
| | Yes | 59 | 95.2 |
| | No | 3 | 4.8 |
| **Ever used a phone alarm in the past 4 weeks** | | | |
| | Yes | 68 | 47.9 |
| | No | 74 | 52.1 |
| **Ever read or sent an SMS in the past 4 weeks** | | | |
| | Yes | 116 | 81.7 |
| | No | 26 | 18.3 |

*n = 62

### Acceptability of SMS reminders

Among AGYW using PrEP in Mukono district, 90.9% (95% CI (84.9, 95.0) accepted use of SMS reminders as an HIV PrEP adherence support tool. Two thirds of the study participants preferred to receive two-way SMS reminders (66.1% (86/130)) and the same proportion preferred to receive daily SMS reminders (66.1% (86/130)). There was no significant difference in the level of acceptability of SMS reminders among those who currently had a mobile phone and those who didn't have (P value 0.720)

### Factors associated with acceptability of SMS reminders

At adjusted analysis, rural residence was negatively associated with acceptability of SMS reminders (aPR 0.92, 95% CI (0.84, 0.99) and believing that SMS cannot breach an individual's privacy (aPR 1.4, 95% CI (1.07, 1.84) was positively associated with acceptability of SMS reminders (Table 3).

## Discussion

This study sought to estimate the acceptability of SMS reminders and the associated factors among AGYW using oral PrEP in Mukono district. Acceptability of SMS reminders among AGYW using PrEP was high with nine out of every 10 AGYW willing to accept SMS. There was also a high ownership of mobile phones by the AGYW. The high level of acceptability of

**Table 3. Unadjusted and adjusted analysis of factors associated with acceptability of SMS reminders among AGYW aged 15–24 years using PrEP in Mukono district.**

| Variable | Acceptability of SMS (n = 129) | cPR (95% CI) | P Value | aPR (95% CI) | P Value |
|---|---|---|---|---|---|
| **Age** | | | | | |
| 15–19 years | 46 (92.0) | 1 | | | |
| 20–24 years | 83 (90.2) | 0.98 (0.88, 1.09) | 0.718 | | |
| **Residence** | | | | | |
| Urban | 39 (97.5) | 1 | | 1 | |
| Rural | 90 (88.2) | 0.91 (0.83, 0.99) | **0.024** | 0.92 (0.84, 0.99) | **0.031** |
| **Period of stay in the current area of residence** | | | | | |
| 5years and above | 52 (89.7) | 1 | | | |
| Less than 5 years | 77 (91.7) | 1.02 (0.92, 1.14) | 0.690 | | |
| **Marital status** | | | | | |
| Married/Cohabiting | 70 (88.6) | 1 | | | |
| Single | 59 (93.7) | 1.06 (0.29, 1.17) | 0.289 | | |
| **Religion** | | | | | |
| Others** | 26 (92.9) | 1 | | | |
| Anglican | 35 (92.1) | 0.99 (0.86, 1.14) | 0.909 | | |
| Catholic | 35 (87.5) | 0.94 (0.81, 1.10) | 0.456 | | |
| Moslem | 33 (91.7) | 0.99 (0.86, 1.14) | 0.859 | | |
| **Occupation** | | | | | |
| Employed | 60 (85.7) | 1 | | | |
| Unemployed | 69 (95.8) | 1.12 (1.00, 1.25) | **0.042** | | |
| **Education level completed** | | | | | |
| Primary level or less | 56 (86.2) | 1 | | | |
| Post-Primary level | 73 (94.8) | 1.10 (0.99, 1.23) | **0.091** | | |
| **Current schooling status** | | | | | |
| Schooling | 37 (92.5) | 1 | | | |
| Not schooling | 92 (90.2) | 0.98 (0.87, 1.09) | 0.651 | | |
| **Moved away from home for 2 consecutive days in past 4 weeks** | | | | | |
| Yes | 24 (85.7) | 1 | | | |
| No | 105(92.1) | 1.08 (0.92, 1.26) | 0.381 | | |
| **Currently own a personal mobile phone** | | | | | |
| Yes | 104(91.2) | 1 | | | |
| No | 25 (89.3) | 0.98 (0.85, 1.13) | 0.765 | | |
| **Share Mobile phone with another person*** | | | | | |
| Yes | 36 (90) | 1 | | | |
| No | 68 (91.9) | 1.02 (0.90, 1.16) | 0.742 | | |
| **Period since PrEP Initiation** | | | | | |
| Past one month | 43 (87.8) | 1 | | | |
| Past two months | 24 (92.3) | 1.05 (0.90, 1.23) | 0.517 | | |
| Past three months or more | 62 (92.5) | 1.06 (0.93, 1.20) | 0.406 | | |
| **Belief that SMS can breach an Individual's privacy** | | | | | |
| Yes | 17 (68.0) | 1 | | 1 | |
| No | 112(95.7) | 1.40 (1.07, 1.85) | **0.014** | 1.40 (1.07, 1.84) | **0.014** |
| **Disclosure of PrEP use** | | | | | |
| Yes | 59 (95.2) | 1 | | | |
| No | 70 (87.5) | 0.92 (0.83, 1.02) | **0.101** | | |
| **Had at least one missed dose of PrEP pills** | | | | | |

(*Continued*)

**Table 3.** (Continued)

| Variable | Acceptability of SMS (n = 129) | cPR (95% CI) | P Value | aPR (95% CI) | P Value |
|---|---|---|---|---|---|
| Yes | 33 (89.2) | 1 | | | |
| No | 96 (91.4) | 1.03 (0.90, 1.16) | 0.702 | | |
| **Ever used a phone alarm in the past 4 weeks** | | | | | |
| Yes | 60 (88.2) | 1 | | | |
| No | 69 (93.2) | 1.06 (0.95, 1.18) | 0.31 | | |
| **Ever read or sent an SMS in the past 4 weeks** | | | | | |
| Yes | 106 91.4) | 1 | | | |
| No | 23 (88.5) | 0.97 (0.83, 1.12) | 0.672 | | |
| **Literacy on use of SMS features of a mobile phone** | | | | | |
| Literate | 119(91.5) | 1 | | | |
| Illiterate | 10 (83.3) | 0.91 (0.70, 1.18) | 0.478 | | |

\* n = 144

\*\*Born again, Seventh Day Adventist

SMS reminders together with high access to mobile phones can be leveraged to support the AGYW to adhere to PrEP through delivery of SMS reminder intervention. Moreover, individuals from rural settings were less likely to accept SMS reminders, while those that believe that SMS reminders cannot breach their privacy were more likely to accept this intervention.

Much as there is paucity of literature that have quantified the level of acceptability of SMS reminders among AGYW taking PrEP, lessons can be drawn from the use of text messages intervention to support adherence among the youth and adult populations on daily medications. For example, the level of acceptability of text messages in the current study is comparable with that reported in a pilot Randomized Controlled Trial (RCT) (88%) involving 202 youth aged 18–22 years conducted in Uganda aimed at assessing the acceptability of a text based program as a primary indication of behavioral change to prevent HIV [19]. This finding is also comparable to a United States of America (USA) pilot RCT assessing the feasibility and acceptability of a daily text messaging intervention designed to promote the well-being, health knowledge and behaviors among 61 youth aged 12–25 year olds over a 16 weeks period [20]. The level of acceptability of SMS reminders in this study was comparable with that reported in a Kenyan study among pregnant women initiated on PrEP where 94% reported that two-way SMS reminders were helpful and 95% reported that they would use them again after the study [21]. Additionally, a qualitative study conducted among trial participants using in-depth interviews following an SMS reminder intervention in Kenya also affirmed that SMS reminders are highly acceptable among AGYW taking PrEP [22]. Another study in central Ethiopia reported the acceptability of SMS reminders among 420 adult PLHIV to be high (91%) [8].

The level of acceptability of SMS reminders in this study is higher than that reported in a cross sectional study (65%) among 100 adolescents aged 12–19 years living with HIV on ART in South Africa [23]. Similarly a cross sectional study involving 801 adults living with HIV in China reported an acceptability of SMS reminders of just 68.9% [9]. The low acceptability seen in these studies compared to this study could be due to the fact that the motivation for taking ART in HIV- negative AGYW as PrEP to prevent HIV may be different from that of PLHIV where the goal is treatment of HIV. Also, in the China study the youth were under represented yet they may have varying levels of acceptability of SMS reminders compared to the adults [9].

In this study, AGYW who stayed in rural settings were 0.92 times less likely to accept the SMS reminders as a PrEP adherence support tool compared to those who stay in urban

settings. This is particularly important in the Ugandan setting where a majority of the population lives in rural areas and nearly two thirds of our study participants were in the rural parts of Mukono district. Rural settings in Uganda occasionally have low levels of socio-economic development compared to urban areas, and often lack well developed telecommunication infrastructure and electricity. This could have far reaching impact on the acceptability of SMS reminder intervention. Therefore AGYW living in such rural settings may require additional support to use this intervention. A cross sectional study conducted in China among 801 adult PLHIV reported that individuals in rural settings were more likely to accept SMS reminders compared to the urban dwellers [9]. This may be due to differences in the level of socio-economic development between the current study settings which is in a low income country compared to China which is a developed nation likely to have well developed telecommunication systems.

The AGYW who believe that SMS cannot breach an individual's privacy were 1.4 times more likely to accept SMS reminders as a PrEP adherence support tool than those who believed that SMS can breach an individual's privacy. Similar findings were reported in a cross sectional study among 420 adult PLHIV in Central Ethiopia [8]. Furthermore, a qualitative study conducted among young PLHIV in Kampala, Uganda, concerns about sharing of phones and accidental disclosure were raised, similar to the findings of this study [13]. In fact, the young people were particularly concerned that in case of breach of confidentiality, they may face stigma and isolation [13]. Breach of individual's privacy may led to unintended disclosure of PrEP use among AGYW which may come with negative consequences like stigma, intimate partner violence, loss of employment and strained social relations among others.

## Limitations

Some data was collected retrospectively and thus prone to recall bias and may have introduced misclassification. However, this was reduced by limiting responses to the past 28 days from date of data collection. Further misclassification could have come from the fact that in this cross sectional study both exposures (independent variables) and outcome were measured at the same time. The study participants were found in their specific safe spaces, however the sample size and data analysis were not adjusted for clustering by these safe spaces and this could have affected internal validity of the study findings, however its impact may be insignificant since the project used the same standard operation procedures in the delivery of the HIV prevention services. A non-probability sampling approach (consecutive sampling) was used to select study participants, thus this may not have been a representative sample much as more than half (58.2%) of the registered PrEP users were involved in the study. The TFA was not previously validated among AGYW to measure acceptability, however it was highly reliable (reliability coefficient was 0.895). Nonetheless, our study reported perceived acceptability of SMS reminders which could be different form the actual acceptability following deployment of the intervention. However, our study provides key information that can guide the packaging of the SMS reminder intervention and subsequent implementation. The study was conducted among AGYW receiving PrEP through the PEPFAR program which provides other incentives like social skills, economic empowerment and business start-ups, and this could limit the generalizability of findings to other AGYW in the general population. Nevertheless, the random error and systematic biases that may have occurred are likely to have had minimal impact on the findings of this study.

## Conclusions

The use of SMS reminders as a PrEP adherence support tool among AGYW was highly acceptable. Having a personal mobile phone and acceptability of SMS reminders were high in this sub-population. SMS reminders can be leveraged to support AGYW to adhere to PrEP but

should be designed in a way that maintains confidentiality, and tailored to supports AGYW living in rural settings.

## Supporting information

**S1 Questionnaire. Study questionnaire.**
(PDF)

## Acknowledgments

We are highly indebted to the research participants for participating in the study. We are especially grateful for the contribution of staff of the Clinical Epidemiology Unit-Makerere University for their invaluable support especially Prof. Joan Kalyango. We appreciate the support from the Makerere University Implementation Science Program especially the Co-Directors Prof. Moses Kamya and Dr. Fred C Semitala, the administrators-Rhoda Namubiru and Moses Ssempala. We thank the management of MUWRP especially the Executive Director-Dr. Hannah Kibuuka, PEPFAR Program Manager–Dr. Fred Magala and Program Officer-Ms. Atim Caroline for the invaluable support during study conduct.

## Author Contributions

**Conceptualization:** Muteebwa Laban, Edith Joloba Nakku, Joan Nangendo, Dan Muramuzi, Sabrina Kitaka Bakeera, Fred Collins Semitala, Aggrey S. Semeere, Charles Karamagi.

**Formal analysis:** Muteebwa Laban, Dan Muramuzi, Faith Akello.

**Funding acquisition:** Fred Collins Semitala.

**Methodology:** Muteebwa Laban, Edith Joloba Nakku, Joan Nangendo, Dan Muramuzi, Faith Akello, Sabrina Kitaka Bakeera, Fred Collins Semitala, Aggrey S. Semeere, Charles Karamagi.

**Project administration:** Muteebwa Laban, Edith Joloba Nakku, Joan Nangendo, Fred Collins Semitala, Aggrey S. Semeere, Charles Karamagi.

**Resources:** Fred Collins Semitala.

**Supervision:** Muteebwa Laban, Edith Joloba Nakku, Joan Nangendo, Sabrina Kitaka Bakeera, Fred Collins Semitala, Aggrey S. Semeere, Charles Karamagi.

**Validation:** Muteebwa Laban.

**Writing – original draft:** Muteebwa Laban, Dan Muramuzi.

**Writing – review & editing:** Muteebwa Laban, Edith Joloba Nakku, Joan Nangendo, Dan Muramuzi, Faith Akello, Sabrina Kitaka Bakeera, Fred Collins Semitala, Aggrey S. Semeere, Charles Karamagi.

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
