## [Decision Letter · Decision Letter 0]

1 Sep 2023

PGPH-D-23-01308

Acceptability of short message service reminders as the support tool for PrEP adherence among young women in Mukono district, Uganda

Dear Dr. Muteebwa,

Thank you for submitting your manuscript to PLOS Global Public Health. After careful consideration, we feel that it has merit but does not fully meet PLOS Global Public Health’s publication criteria as it currently stands. Therefore, we invite you to submit a revised version of the manuscript that addresses the points raised during the review process.

We look forward to receiving your revised manuscript.

Kind regards,

Henry Zakumumpa, PhD

Academic Editor

Journal Requirements:

Additional Editor Comments (if provided):

We are pleased to share comments from our reviewers. Please pay attention to the discussion section and endevour to strengthen in line with comments from the reviewers. Also, please indicate a line on the exclusion criteria for your study.

Reviewers' comments:

Reviewer's Responses to Questions

**Comments to the Author**

1. Does this manuscript meet PLOS Global Public Health’s publication criteria? Is the manuscript technically sound, and do the data support the conclusions? The manuscript must describe methodologically and ethically rigorous research with conclusions that are appropriately drawn based on the data presented.

Reviewer #1: Yes

Reviewer #2: Yes

2. Has the statistical analysis been performed appropriately and rigorously?

Reviewer #1: Yes

Reviewer #2: Yes

3. Have the authors made all data underlying the findings in their manuscript fully available (please refer to the Data Availability Statement at the start of the manuscript PDF file)?

Reviewer #1: Yes

Reviewer #2: Yes

4. Is the manuscript presented in an intelligible fashion and written in standard English?

Reviewer #1: Yes

Reviewer #2: Yes

5. Review Comments to the Author

Reviewer #1: Although technology-based interventions have been recommended to support the effective use of daily oral PrEP among AGYW, few studies have assessed the acceptability of technology-based interventions such as SMS among AGYW. The study includes relevant information to fill this existing gap. The authors can consider the following items:

Introduction

• Line 47- suggest removing “taking PrEP” as already stated in the sentence.

• Lines 60-61: the data provided is from 2014. Any more up-to-date data?

Methods

• Was this cross-sectional study conducted in DREAMS-supported sites? If so, suggest stating this information to provide additional context.

• Inclusion criteria- specify if you are referring to at least one month of daily oral PrEP (versus stating PrEP alone).

• Suggest including more information on how AGYW were recruited to complete the survey. Was it a convenient sample of AGYW from across a specific number of DREAMS supported sites or safe places? Please provide further information on how AGYW were recruited.

• Please provide more information about the acceptability of SMS reminders. Did the researchers use a validated scale? If so, specify and state if previously used to assess the acceptability of technology-based interventions for PrEP.

• Line 96: “Participants with a score above 21 were categorized as having accepted SMS reminders”. Please provide information for selecting this specific threshold.

• Line 103- “We analyzed data in STATA…”. Suggest including this sentence as the last sentence of the data analysis.

• Lines 109- “We used the modified Poisson regression with robust standard of errors”. The measure of effect is prevalence ratio. Any reasons for not using multivariable log-binomial regression model?

Results

• Suggest removing the term “slightly” when presenting the results.

• Lines 137 “Of those who owned a mobile phone, almost two thirds reported that they share it with another person (64.9% (74/114).” This information contradicts the values in the table. Revise one of them for consistency.

• Suggest removing the the number of individualsin the title of your table. This information is included in the tables.

• Tables- for consistency, use the same number of decimals (e.g.2) for all numbers

Discussion

• General comments on the discussion: Suggest revising the discussion to compare/contrast your results to other studies related to PrEP among AGYW versus expanding HIV treatment among different populations (as much as possible). Currently, the discussion includes few references, and many of them are not directly related to PrEP, technology-based interventions and AGYW.

• Paragraph 206-208- suggest rephrasing or removing this paragraph as it does not add to the study and also claims are not well supported by the scientific literature.

• Suggest reviewing the limitations. Some additional limitations to consider (1) the study focuses on the perceived acceptability of SMS which may differ from the acceptability in real-life once AGYW receive SMS for PrEP support; (2) the survey seems to have recruited a convenient sample of AGYW (not randomly selected); and (3) the survey did not seems to have used validated scale among the study population.

Reviewer #2: Overall, the paper gives great insights into SMS acceptability as a support tool for PrEP use among young women. The author may consider the following comments;

Line 30-31: Please specify the direction of the association of these two variables.

Line 57: It may be important for the reader to know why AGYW has increased levels of stigma compared to other populations.

The author states in the abstract that TFA was used, may be important to mention briefly in the background what it is and why it was important and relevant to use it when introducing your study in the final paragraph.

Line 69: The author should consider providing citations for these statements: The district has a population of 599,817 people of whom about 187,143 are aged 10 to 24 years. The HIV prevalence among persons aged 15 years and above in Mukono district is 6.1% which is higher than the national prevalence (5.4%) thus is a priority district for HIV prevention.

Line 81: Saying the study had no exclusion criteria may mean anyone could possibly participate in the study yet you outline the inclusion criteria. The author may consider stating that individuals who did not meet these criteria were excluded.

Line 93: Consider attaching the questionnaire as a supplement tool or mention the kind of question the participants were asked for the reader to understand.

In the results sections, ideally, the second table should show the main findings of your study. Consider providing a table or figure that shows acceptability. Although the reported individual characteristics are important, it is essential to tell the reader your key findings early in your result section.

Line 164-166: On factors associated with acceptability, the author just mentions that there is an association, it is important to give the direction of the association is it positive or negative for clarity. Is it increasing the likelihood of acceptability of SMS or is it reducing the likelihood? Also, consider stating that there were no other variables associated with the SMS acceptability.

Line 169: The asterick in Table 3 doesn’t appear anywhere in the table. Also, is there an explanation why (n) in the acceptability result is different with the (n) in your demographics?

In the discussion, the author has done well in reporting other studies and reviewing literature but has failed to give strength to their own findings. First, the first paragraph in your discussion does not fully summarize your findings. Second, it’s important to be clear on what you found and how it is relevant to other studies. There are different significant aspects of your findings that are not clear in your discussion. For instance, there was a significant association between rural areas and breach of privacy how does that compare with other literature? There were also preferences in either one or two-way SMS how does that compare?

Line 188: In this study, the behavioral changes they targeted didn’t involve a lot of stigma like that associated with the use of PrEP. What does this mean and what behavioral changes are implied here?

Line 201 Motivation for taking ART in HIV-negative AGYW may be confusing to the reader the author may consider just calling it PrEP.

Line 203: Also, in the China study the youth were underrepresented yet they may have varying levels of acceptability of SMS reminders compared to the adults. This statement is unclear, how were they underrepresented? and in which context? Also, provide a citation.

Line 212: It would only be fair to compare with the same population I think adults’ level of acceptability may differ from AGYW.

Line 206: Specify the value in % for clarity.

Paragraph 206: The author states that in Uganda most population are from rural area, What is the implication of this given that the rural area is less likely to accept SMS? Consider reviewing studies showing that SMS helps in adherence and how that may have a potential impact on rural residents.

Line 213: Clarify how social economic differences impact either acceptance or none acceptance of the SMS

Paragraph 217: Consider reviewing studies that have assessed the outcome of breach of confidentiality in PrEP use especially among young women and not just PLHIV.

Line 224: It needs to come out clearly in your method section that some data were collected retrospectively.

6. PLOS authors have the option to publish the peer review history of their article (what does this mean?). If published, this will include your full peer review and any attached files.

**Do you want your identity to be public for this peer review?** For information about this choice, including consent withdrawal, please see our Privacy Policy.

Reviewer #1: No

Reviewer #2: No

---

## [Editor Report · Decision Letter 1]

31 Oct 2023

PGPH-D-23-01308R1

Acceptability of short message service reminders as the support tool for PrEP adherence among young women in Mukono district, Uganda

Dear Dr Muheebwa Laban,

Thank you for submitting your manuscript to PLOS Global Public Health. After careful consideration, we feel that it has merit but does not fully meet PLOS Global Public Health’s publication criteria as it currently stands. Therefore, we invite you to submit a revised version of the manuscript that addresses the points raised during the review process.

We look forward to receiving your revised manuscript.

Kind regards,

Henry Zakumumpa, PhD

Academic Editor

Journal Requirements:

Additional Editor Comments (if provided):

We are pleased to share reports from our two reviewers. I point you two comments they have raised. There is need to do more to acknowledge your study limitations. Please add more detail to the inclusion criteria for your study.

One of the reviewers has requested that you attach the study questionnaire.
---

## [Editor Report · Decision Letter 2]

21 Nov 2023

Acceptability of short message service reminders as the support tool for PrEP adherence among young women in Mukono district, Uganda

PGPH-D-23-01308R2

Dear Laban Muteebwa,

We are pleased to inform you that your manuscript 'Acceptability of short message service reminders as the support tool for PrEP adherence among young women in Mukono district, Uganda' has been provisionally accepted for publication in PLOS Global Public Health.

Best regards,

Henry Zakumumpa, PhD

Academic Editor